# A COMPREHENSIVE, APPLICATION-ORIENTED STUDY OF CATASTROPHIC FORGETTING IN DNNs

**B. Pfülb & A. Gepperth**
Department of Computer Science
Hochschule Fulda
Fulda 36037, Germany
{benedikt.pfuelb,alexander.gepperth}@cs.hs-fulda.de

## ABSTRACT

We present a large-scale empirical study of catastrophic forgetting (CF) in modern Deep Neural Network (DNN) models that perform sequential (or: incremental) learning. A new experimental protocol is proposed that enforces typical constraints encountered in application scenarios. As the investigation is empirical, we evaluate CF behavior on the hitherto largest number of visual classification datasets, from each of which we construct a representative number of Sequential Learning Tasks (SLTs) in close alignment to previous works on CF. Our results clearly indicate that there is no model that avoids CF for all investigated datasets and SLTs under application conditions. We conclude with a discussion of potential solutions and workarounds to CF, notably for the EWC and IMM models.

## 1 INTRODUCTION

This article is in the context of sequential or incremental learning in Deep Neural Networks (DNNs). Essentially, this means that a DNN is not trained once, on a single task $D$, but successively on two or more sub-tasks $D_1, \ldots, D_n$, one after another. Learning tasks of this type, which we term Sequential Learning Tasks (SLTs) (see Fig. 1a), are potentially very common in real-world applications. They occur wherever DNNs need to update their capabilities on-site and over time: gesture recognition, network traffic analysis, or face and object recognition in mobile robots. In such scenarios, neural networks have long been known to suffer from a problem termed "catastrophic forgetting"(CF) (e.g., French (1999)) which denotes the abrupt and near-complete loss of knowledge from previous sub-tasks $D_1, \ldots, D_{k-1}$ after only a few training iterations on the current sub-task $D_k$ (see Fig. 1b compared to Fig. 1c). We focus on SLTs from the visual domain with two sub-tasks each, as DNNs show pronounced CF behavior even when only two sub-tasks are involved.

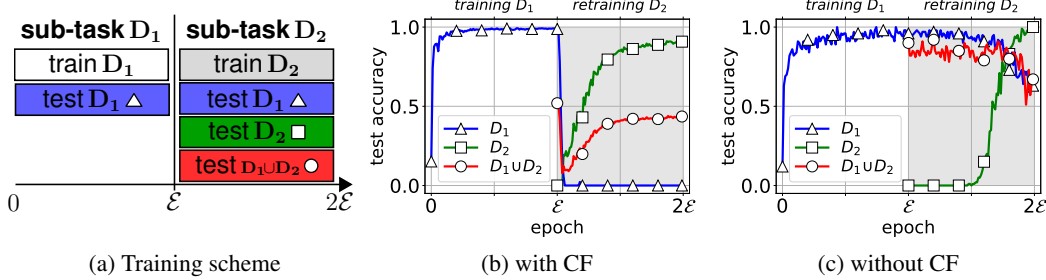

(a) Training scheme      (b) with CF      (c) without CF

Figure 1: Scheme of incremental training experiments conducted in this article (a) and representative outcomes with (b) and without CF (c). The sequential learning tasks used in this study only have two sub-tasks: $D_1$ and $D_2$. During training (white background) and re-training (gray background), test accuracy is measured on $D_1$ (blue, △), $D_2$ (green, □) and $D_1 \cup D_2$ (red, ○). The blue curve allows to determine the presence of CF by simple visual inspection: if there is significant degradation w.r.t. the red curve, then CF has occurred.

## 1.1 DISCUSSION OF RELATED WORK ON CF

The field of incremental learning is large, e.g., Parisi et al. (2018) and Gepperth & Hammer (2016). Recent systematic comparisons between different DNN approaches to avoid CF are performed in, e.g., Serra et al. (2018) or Kemker et al. (2018). Principal recent approaches to avoid CF include ensemble methods (Ren et al., 2017; Fernando et al., 2017), dual-memory systems (Shin et al., 2017; Kemker & Kanan, 2017; Rebuffi et al., 2017; Gepperth & Karaoguz, 2016) and regularization approaches. Whereas Goodfellow et al. (2013) suggest Dropout for alleviating CF, the EWC method (Kirkpatrick et al., 2017) proposes to add a term to the energy function that protects weights that are important for the previous sub-task(s). Importance is determined by approximating the Fisher information matrix of the DNN. A related approach is pursued by the Incremental Moment Matching technique (IMM) (see Lee et al. (2017)), where weights from DNNs trained on a current and a past sub-tasks are "merged" using the Fisher information matrix. Other regularization-oriented approaches are proposed in Aljundi et al. (2018); Srivastava et al. (2013) and Kim et al. (2018) which focus on enforcing sparsity of neural activities by lateral interactions within a layer.

**Number of tested datasets** In general, most methods referenced here are evaluated only on a few datasets, usually on MNIST (LeCun et al., 1998) and various derivations thereof (permutation, rotation, class separation). Some studies make limited use of CIFAR10, SVHN, the Amazon sentiment analysis problem, and non-visual problems such as data from Q-learning of Atari games. A large-scale evaluation on a huge number of qualitatively different datasets is still missing[1].

**Model selection and prescience** Model selection (i.e., selecting DNN topology and hyper-parameters) is addressed in some approaches (Goodfellow et al., 2013) but on the basis of a "prescient" evaluation where the best model is selected *after* all tasks have been processed, an approach which is replicated in Kirkpatrick et al. (2017). This amounts to a knowledge of future sub-tasks which is problematic in applications. Most approaches ignore model selection (Lee et al., 2017; Srivastava et al., 2013; Aljundi et al., 2018; Kim et al., 2018), and thus implicitly violate causality.

**Storage of data from previous sub-tasks** From a technical point of view, DNNs can be retrained without storing training data from previous sub-tasks, which is done in Goodfellow et al. (2013) and Srivastava et al. (2013). For regularization approaches, however, there are regularization parameters that control the retention of previous knowledge, and thus must be chosen with care. In Kirkpatrick et al. (2017), this is $\lambda$, whereas two such quantities occur in Lee et al. (2017): the "balancing" parameter $\alpha$ and the regularization parameter $\lambda$ for L2-transfer. The only study where regularization parameters are obtained through cross-validation (which is avoided in other studies) is Aljundi et al. (2018) (for $\lambda_{SNI}$ and $\lambda_{\Omega}$) but this requires to store all previous training data.

This review shows that enormous progress has been made, but that there are shortcomings tied to applied scenarios which need to be addressed. We will formalize this in Sec. 1.2 and propose an evaluation strategy that takes these formal constraints into account when testing CF in DNNs.

## 1.2 INCREMENTAL LEARNING IN APPLIED SCENARIOS

When training a DNN model on SLTs, first of all the model must be able to be retrained at any time by new classes (class-incremental learning). Secondly, it must exhibit retention, or at least graceful decay, of performance on previously trained classes. Some forgetting is probably unavoidable, but it should be gradual and not immediate, i.e., catastrophic. However, if a DNN is operating in, e.g., embedded devices or autonomous robots, additional conditions may be applicable:

**Low memory footprint** Data from past sub-tasks cannot be stored and used for re-training, or else to determine when to stop re-training.

**Causality** Data from future sub-tasks, which are often known in academic studies but not in applications, must not be utilized in any way, especially not for DNN model selection. This point might seem trivial, but a number of studies such as Kirkpatrick et al. (2017); Goodfellow et al. (2013) and Srivastava et al. (2013) perform model selection in hindsight, after having processed all sub-tasks.

**Constant update complexity** Re-training complexity (time and memory) must not depend on the number of previous sub-tasks, thus more or less excluding replay-based schemes such as Shin et al. (2017). Clearly, even if update complexity is constant w.r.t. the number of previous sub-tasks, it should not be too high in absolute terms either.

---

[1]Although the comparisons performed in Aljundi et al. (2018) include many datasets, the experimental protocol is unclear, so it is uncertain how to interpret these results.

### 1.3    Contribution and principal conclusions

The original contributions of our work can be summarized as follows:

- We propose a training and evaluation paradigm for incremental learning in DNNs that enforces typical application constraints, see Sec. 1.2. The importance of such an application-oriented paradigm is underlined by the fact that taking application constraints into account leads to radically different conclusions about CF than those obtained by other recent studies on CF (see Sec. 1.1).

- We investigate the incremental learning capacity of various DNN approaches (Dropout, LWTA, EWC and IMM) using the largest number of qualitatively different classification datasets so far described. We find that all investigated models are afflicted by catastrophic forgetting, or else in violation of application constraints and discuss potential workarounds.

- We establish that the "permuted" type of SLTs (e.g., "permuted MNIST") should be used with caution when testing for CF.

- We do **not** propose a method for avoiding CF in this article. This is because avoiding CF requires a consensus on how to actually measure this effect: our novel contribution is a proposal how to do just that.

## 2    Methods and DNN models

We collect a large number of visual classification datasets, from each of which we construct SLTs according to a common scheme, and compare several recent DNN models using these SLTs. The experimental protocol is such that application constraints, see Sec. 1.2, are enforced. For all tested DNN models (see below), we use a TensorFlow (v1.7) implementation under Python (v3.4 and later). The source code for all processed models, the experiment-generator and evaluation routine can be found on our public available repository [2].

**FC** A normal, fully-connected (FC) feed-forward DNN with a variable number and size of hidden layers, each followed by ReLU, and a softmax readout layer minimizing cross-entropy.
**CONV** A convolutional neural network (CNN) based on the work of Cirean et al. (2011). It is optimized to perform well on image classification problems like MNIST. We use a fixed topology: two conv-layers with $32$ and $64$ filters of size $5 \times 5$ plus ReLU and $2 \times 2$ max-pooling, followed by a fc-layer with $1024$ neurons and softmax readout layer minimizing a cross-entropy energy function.
**EWC** The Elastic Weight Consolidation (EWC) model presented by Kirkpatrick et al. (2017).
**LWTA** A fully-connected DNN with a variable number and size of hidden layers, each followed by a Local Winner Takes All (LWTA) transfer function as proposed in Srivastava et al. (2013).
**IMM** The Incremental Moment Matching model as presented by Lee et al. (2017). We examine the weight-transfer techniques in our experiments, using the provided implementation.
**D-FC and D-CONV** Motivated by Goodfellow et al. (2013) we combine the FC and CONV models with Dropout as an approach to solve the CF problem. Only FC and CONV are eligible for this, as EWC and IMM include dropout by default, and LWTA is incompatible with Dropout.

### 2.1    Hyper-parameters and model selection

We perform model selection in all our experiments by a combinatorial hyper-parameter optimization, whose limits are imposed by the computational resources available for this study. In particular, we vary the number of hidden layers $L \in \{2, 3\}$ and their size $S \in \{200, 400, 800\}$ (CNNs excluded), the learning rate $\epsilon_1 \in \{0.01, 0.001\}$ for sub-task $D_1$, and the re-training learning rate $\epsilon_2 \in \{0.001, 0.0001, 0.00001\}$ for sub-task $D_2$. The batch size (batch$_{size}$) is fixed to 100 for all experiments, and is used for both training and testing. As in other studies, we do not use a fixed number of training iterations, but specify the number of training *epochs* (i.e., passes through the whole dataset) as $\mathcal{E} = 10$ for each processed dataset (see Sec. 2.2), which allows an approximate comparison of different datasets. The number of training/testing batches per epoch, $\mathcal{B}$, can be calculated from the batch size and the currently used dataset size. The set of all hyper-parameters for

---

[2]`https://gitlab.informatik.hs-fulda.de/ML-Projects/CF_in_DNNs`

a certain model, denoted $\mathcal{P}$, is formed as a Cartesian product from the allowed values of the hyper-parameters $L$, $S$, $\epsilon_1$, $\epsilon_2$ and complemented by hyper-parameters that remain fixed ($\mathcal{E}$, batch$_{\text{size}}$) or are particular to a certain model. For all models that use dropout, the dropout rate for the input layer is fixed to $0.2$, and to $0.5$ for all hidden layers. For CNNs, the dropout rate is set to $0.5$ for both input and hidden layers. All other hyper-parameters for CNNs are fixed, e.g., number and size of layers, the max-pooling and filter sizes and the strides ($2 \times 2$) for each channel. These decisions were made based on the work of Goodfellow et al. (2013). The LWTA block size is fixed to $2$, based on the work of Srivastava et al. (2013). The model parameter $\lambda$ for EWC is set to $\lambda 1/\epsilon_2$ (set but not described in the source code of Kirkpatrick et al. (2017)). For all models except IMM, the momentum parameter for the optimizer is set to $\mu = 0.99$ (Sutskever et al., 2013). For the IMM models, the SGD optimizer is used, and the regularizer value for the L2-regularization is set to $0.01$ for L2-transfer and to $0.0$ for weight transfer.

## 2.2 Datasets

We select the following datasets (see Tab. 1). In order to construct SLTs uniformly across datasets, we choose the 10 best-represented classes (or random classes if balanced) if more are present.

**MNIST** (LeCun et al., 1998) is the common benchmark for computer vision systems and classification problems. It consist of gray scale images of handwritten digits (0-9).
**EMNIST** (Cohen et al., 2017) is an extended version of MNIST with additional classes of handwritten letters. There are different variations of this dataset: we extract the ten best-represented classes from the *By_Class* variation containing 62 classes.
**Fruits 360** (Murean & Oltean, 2017) is a dataset comprising fruit color images from different rotation angles spread over 75 classes, from which we extract the ten best-represented ones.
**Devanagari** (Acharya et al., 2015) contains gray scale images of Devanagari handwritten letters. From the 46 character classes (1.700 images per class) we extract 10 random classes.
**FashionMNIST** (Xiao et al., 2017) consists of images of clothes in 10 classes and is structured like the MNIST dataset. We use this dataset for our investigations because it is a "more challenging classification task than the simple MNIST digits data (Xiao et al., 2017)".
**SVHN** (Netzer et al., 2011) is a 10-class dataset based on photos of house numbers (0-9). We use the cropped digit format, where the number is centered in the color image.
**CIFAR10** (Krizhevsky, 2009) contains color images of real-world objects e.g, dogs, airplanes etc.
**NotMNIST** (Bulatov Yaroslav) contains grayscale images of the 10 letter classes from "A" to "J", taken from different publicly available fonts.
**MADBase** (Abdelazeem Sherif & El-Sherif Ezzat) is a modified version of the "Arabic Digits dataBase", containing grayscale images of handwritten digits written by 700 different persons.

Table 1: Overview of each dataset's detailed properties. Image dimensions are given as width $\times$ height $\times$ channels. Concerning data imbalance, the largest percentual difference in sample count between any two classes is given for training and test data, a value of 0 indicating a perfectly balanced dataset.

| Properties \\ Dataset | image size | number of elements | | class balance (%) | |
|---|---|---|---|---|---|
| | | train | test | train | test |
| CIFAR10 | $32 \times 32 \times 3$ | 50.000 | 10.000 | 0 | 0 |
| Devanagari | $32 \times 32 \times 1$ | 18.000 | 2.000 | 0.3 | 2.7 |
| EMNIST | $28 \times 28 \times 1$ | 345.035 | 57.918 | 2.0 | 2.0 |
| FashionMNIST | $28 \times 28 \times 1$ | 60.000 | 10.000 | 0 | 0 |
| Fruits 360 | $100 \times 100 \times 3$ | 6.148 | 2.052 | 4.0 | 4.2 |
| MADBase | $28 \times 28 \times 1$ | 60.000 | 10.000 | 0 | 0 |
| MNIST | $28 \times 28 \times 1$ | 55.000 | 10.000 | 2.2 | 2.4 |
| NotMNIST | $28 \times 28 \times 1$ | 529.114 | 18.724 | $\sim 0$ | $\sim 0$ |
| SVHN | $32 \times 32 \times 3$ | 73.257 | 26.032 | 12.6 | 13.5 |

## 2.3 Sequential Learning Tasks (SLTs)

As described in Sec. 1, each SLT consists of two sub-tasks $D_1$ and $D_2$. For each dataset (see Sec. 2.2), these are defined by either applying different spatial permutations to all image data (DP10-

10 type SLTs), or by subdividing classes into disjunct groups (see Tab. 2). For the latter case, we include SLTs where the second sub-task adds only 1 class (D9-1 type SLTs) or 5 classes (D5-5 type SLTs), since CF may be tied to how much newness is introduced. We include permutations (DP10-10) since we suspected that this type of SLT is somehow much easier than others, and therefore not a good incremental learning benchmark. As there are far more ways to create D5-5 type SLTs than D9-1 type SLTs, we create more of the former (8-vs-3) in order to avoid misleading results due to a particular choice of subdivision, whereas we create only a single permutation-type SLT.

Table 2: Overview of all SLTs. The assignment of classes to sub-tasks $D_1$ and $D_2$ are disjunct, except for DP10-10 where two different seeded random image permutations are applied.

| $\overset{\text{SLT}}{\rightarrow}$ | D5-5a | D5-5b | D5-5c | D5-5d | D5-5e | D5-5f | D5-5g | D5-5h | D9-1a | D9-1b | D9-1c | DP10-10 |
|---|---|---|---|---|---|---|---|---|---|---|---|---|
| $D_1$ | 0-4 | 02468 | 34689 | 02567 | 01345 | 03489 | 05678 | 02368 | 0-8 | 1-9 | 02-9 | 0-9 |
| $D_2$ | 5-9 | 13579 | 01257 | 13489 | 26789 | 12567 | 12349 | 14579 | 9 | 0 | 1 | 0-9 |

## 3 EXPERIMENTS

This study presents just one, albeit very large, experiment, whose experimental protocol implements the constraints from Sec. 1.2.

Every DNN model from Sec. 2 is applied to each SLT as defined in Sec. 2.3 while taking into account model selection, see Sec. 2.1. A precise definition of our application-oriented experimental protocol is given in Alg. 1. For a given model $m$ and an SLT ($D_1$ and $D_2$), the first step is to determine the best hyper-parameter vector $\vec{p}^*$ for sub-task $D_1$ only (see lines 1-4), which determines the model $m_{\vec{p}^*}$ used for re-training. In a second step, $m_{\vec{p}^*}$ (from line 5) is used for re-training on $D_2$, with a different learning rate $\epsilon_2$ which is varied separately. We introduce two criteria for determining the ($\epsilon_2$-dependent) quality of a re-training phase (lines 6-10): "best", defined by the highest test accuracy on $D_1 \cup D_2$, and "last", defined by the test accuracy on $D_1 \cup D_2$ at the end of re-training. Although the "best" criterion violates the application constraints of Sec. 1.2 (requires $D_1$), we include it for comparison purposes. Finally, the result is computed as the highest $\epsilon_2$-dependent quality (line 11). Independently of the second step, another training of $m_{\vec{p}^*}$ is conducted using $D_1 \cup D_2$, resulting in what we term *baseline accuracy*.

Evaluation for IMM differs slightly: in line 5, a copy of $m_{\vec{p}^*}$ is kept, termed $m_{\vec{p}^*}^1$, and the weights of $m_{\vec{p}^*}$ are re-initialized. After selecting the best re-trained model $m_{\vec{p}^*}^2$ as a function of $\epsilon_2$, final performance $q_{\vec{p}^*}$ is obtained by "merging" the models $m_{\vec{p}^*}^1$ and $m_{\vec{p}^*}^2$ and testing the result.

---

**Algorithm 1:** The application-oriented evaluation strategy used in this study.

---
**Data:** model $m$, SLT with sub-tasks $D_1$, $D_2$, hyper-parameter value set $\mathcal{P}$
**Result:** incremental learning quality for model with hyper-parameters $\vec{p}^*$: $q_{\vec{p}^*}$

1  **forall** $\vec{p} \in \mathcal{P}$ **do**          // determine accuracy for all hyper-parameters when training on $D_1$
2  $\quad$ **for** $t \leftarrow 0$ **to** $\mathcal{E} \cdot \mathcal{B}$ **do**
3  $\quad\quad$ train($m_{\vec{p}}, D_1^{train}, \epsilon_1$)
4  $\quad\quad$ $q_{\vec{p},t} \leftarrow$ test($m_{\vec{p}}, D_1^{test}, t$)

5  $m_{\vec{p}^*} \leftarrow$ model $m_{\vec{p},t}$ with maximum $q_{\vec{p},t}$          // find best model with max. accuracy on $D_1$
6  **forall** $\epsilon_2$ **do**                                                                  // find best $\epsilon_2$ value
7  $\quad$ $m_{\vec{p}^*,\epsilon_2} \leftarrow m_{\vec{p}^*}$
8  $\quad$ **for** $t \leftarrow 0$ **to** $\mathcal{E} \cdot \mathcal{B}$ **do**
9  $\quad\quad$ train($m_{\vec{p}^*,\epsilon_2}, D_2^{train}, \epsilon_2$)
10 $\quad\quad$ $q_{\vec{p}^*,t,\epsilon_2} \leftarrow$ test($m_{\vec{p}^*,\epsilon_2}, D_2^{test}, t$)

11 $q_{\vec{p}^*} \leftarrow \max_{\epsilon_2}$ best/last$_t$ $q_{\vec{p},t,\epsilon_2}$          // find parameter set with the best accuracy on $D_2$

---

## 4 FINDINGS

The results of the experiment described in Sec. 3 are summarized in Tab. 3, and in Tab. 4 for IMM. They lead us to the following principal conclusions:

**Permutation-based SLTs should be used with caution** We find that DP10-10, the SLT based on permutation, does not show CF for **any** model and dataset, which is exemplary visualized for the FC model in Fig. 2 which fails completely for all other SLTs. While we show this only for SLTs with two sub-tasks, and it is unknown how experiments with more sub-tasks would turn out, we nevertheless suggest caution when intepreting results on permutation-based SLTs.

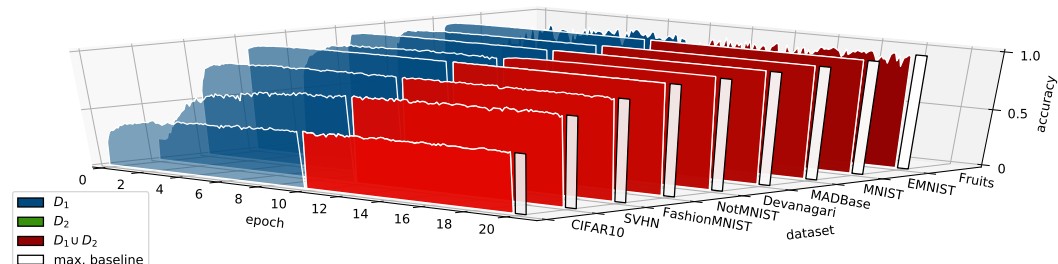

Figure 2: Best FC experiments for DP10-10. The blue surfaces (epochs 0-10) represent the accuracy on $D_1$, the green (covered here) and red surfaces the accuracy on $D_2$ and $D_1 \cup D_2$ during re-training (epochs 10-20). The white bars indicate baseline performance. See also Appendix B for 2D plots.

**All examined models exhibit CF** While this is not surprising for FC and CONV, D-FC as proposed in Goodfellow et al. (2013) performs poorly (see Fig. 3), as does LWTA (Srivastava et al., 2013). For EWC and IMM, the story is slightly more complex and will be discussed below.

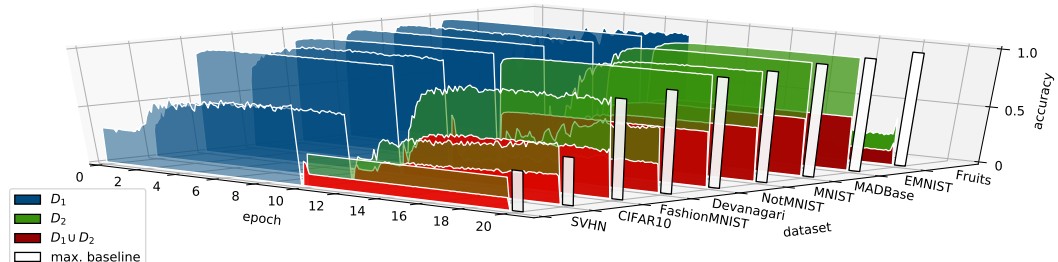

Figure 3: Best D-FC experiments for SLT D5-5, to be read as Fig. 2 and showing the occurrence of CF. See also Appendix B for 2D plots.

**EWC is mildly effective against CF for simple SLTs.** Our experiments shows that EWC is effective against CF for D9-1 type SLTs, at least when the "best" evaluation criterion is used, which makes use of $D_1$. This, in turn, violates the application requirements of Sec. 1.2. For the "last" criterion not making use of $D_1$, EWC performance, though still significant, is much less impressive. We can see the origins of this difference illustrated in Fig. 4.

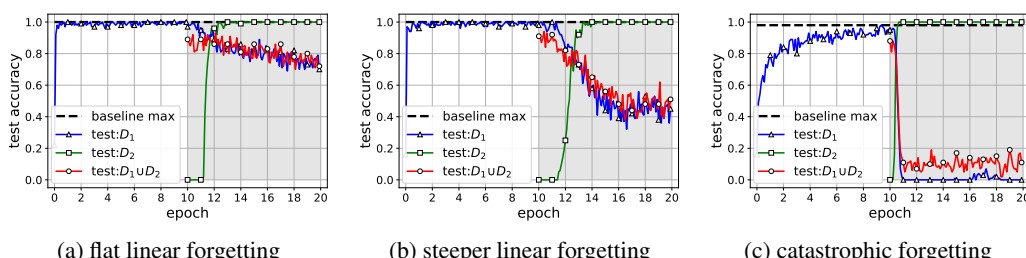

(a) flat linear forgetting     (b) steeper linear forgetting     (c) catastrophic forgetting

Figure 4: Illustrating the difference between the "best" and "last" criterion for EWC. Shown is the accuracy over time for the best model on SLT D9-1c using EMNIST (a), D9-1a using EMNIST (b) and D9-1b using Devanagari (c). The blue curve ($\triangle$) measures the accuracy on $D_1$, green ($\square$) only on $D_2$ and red ($\bigcirc$) the $D_1 \cup D_2$ during the training (white) and the re-training phase (gray). Additionally, the baseline (dashed line) is indicated. In all three experiments, the "best" strategy results in approximately 90% accuracy, occurring at the beginning of re-training when $D_2$ has not been learned yet. Here, the magnitude of the best/last difference is a good indicator of CF which clearly happens in (c), partly in (b) and slightly or not at all in (a).

**EWC is ineffective against CF for more complex problems.** Tab. 3 shows that EWC cannot prevent CF for D5-5 type SLTs, see Fig. 5. Apparently, the EWC mechanism cannot protect all the weights relevant for $D_1$ here, which is likely to be connected to the fact that the number of samples in both sub-tasks is similar. This is not the case for D9-1 type tasks where EWC does better and where $D_2$ has about $10\%$ of the samples in $D_1$.

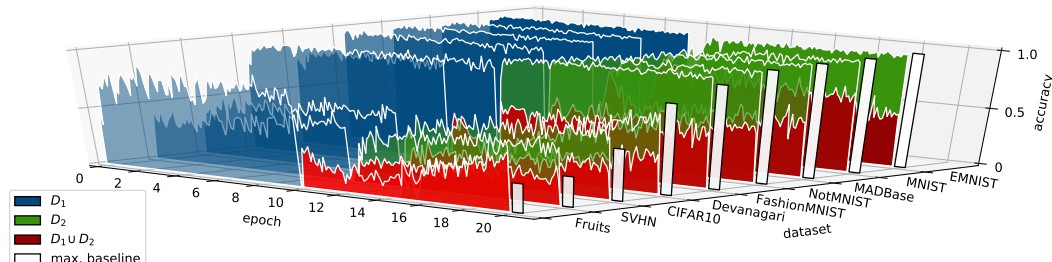

Figure 5: Best EWC experiments for SLT D5-5d constructed from all datasets, to be read as Fig. 2. We observe that CF happens for all datasets. See also Appendix B for 2D plots.

Table 3: Summary of incremental learning quality $q_{\tilde{p}*}$, see Alg. 1, over SLTs of type D9-1, D5-5 and DP10-10 (see also Tab. 5). For aggregating results over SLTs of the same type, the minimal value of $q_{\tilde{p}*}$ is taken. Each cell contains two qualities evaluated according to the "best" and "last" criteria, see Alg. 1. Cell coloring was determined according to "best". For DP10-10 and D5-5 type tasks, CF (black cells) is indicated by qualities $< 0.5$. The corresponding threshold for D9-1 type tasks is $0.9$. Only when the threshold is exceeded, re-training can be regarded as successful which is visualized by a grayscale gradient (black – gray – white).

| DS | SLT | FC | D-FC | CONV | D-CONV | LWTA | EWC |
|---|---|---|---|---|---|---|---|
| CIFAR10 | D5-5 | .30/.28 | .26/.23 | .31/.10 | .30/.18 | .31/.30 | .32/.20 |
| | D9-1 | .45/.10 | .37/.10 | .45/.10 | .48/.10 | .45/.10 | .36/.08 |
| | DP10-10 | .54/.52 | .44/.43 | .52/.50 | .56/.55 | .54/.51 | .57/.46 |
| Devanagari | D5-5 | .49/.42 | .46/.26 | .49/.45 | .49/.11 | .11/.10 | .40/.23 |
| | D9-1 | .86/.10 | .84/.09 | .88/.10 | .89/.09 | .86/.09 | .88/.09 |
| | DP10-10 | .98/.98 | .98/.98 | .95/.95 | 1.0/1.0 | .97/.96 | 1.0/.96 |
| EMNIST | D5-5 | .50/.48 | .50/.48 | .50/.48 | .50/.48 | .50/.48 | .36/.08 |
| | D9-1 | .88/.09 | .88/.09 | .89/.09 | .89/.09 | .88/.09 | .92/.51 |
| | DP10-10 | .99/.99 | .99/.99 | 1.0/1.0 | 1.0/1.0 | .99/.99 | 1.0/.98 |
| F-MNIST | D5-5 | .46/.45 | .46/.44 | .47/.45 | .46/.46 | .46/.46 | .55/.47 |
| | D9-1 | .78/.10 | .77/.10 | .81/.10 | .81/.10 | .78/.10 | .85/.50 |
| | DP10-10 | .90/.88 | .88/.87 | .92/.92 | .92/.92 | .90/.89 | .95/.95 |
| Fruits | D5-5 | .32/.14 | .46/.13 | .14/.09 | .14/.09 | .28/.11 | .34/.03 |
| | D9-1 | .34/.09 | .38/.21 | .14/.09 | .23/.09 | .38/.09 | .55/.13 |
| | DP10-10 | 1.0/.97 | 1.0/.99 | .90/.88 | .97/.96 | .98/.12 | .98/.90 |
| MADBase | D5-5 | .49/.49 | .49/.49 | .49/.49 | .49/.10 | .50/.49 | .40/.26 |
| | D9-1 | .89/.10 | .91/.10 | .89/.10 | .90/.10 | .94/.10 | .99/.70 |
| | DP10-10 | .99/.99 | .99/.99 | .99/.99 | .99/.99 | .99/.98 | 1.0/.99 |
| MNIST | D5-5 | .49/.48 | .49/.47 | .48/.11 | .48/.15 | .10/.09 | .50/.31 |
| | D9-1 | .88/.10 | .88/.10 | .88/.10 | .88/.10 | .87/.10 | .99/.71 |
| | DP10-10 | .99/.99 | .98/.98 | .99/.99 | .99/.99 | .98/.98 | 1.0/.98 |
| NotMNIST | D5-5 | .49/.49 | .49/.49 | .49/.49 | .50/.49 | .50/.49 | .57/.50 |
| | D9-1 | .87/.10 | .86/.10 | .88/.10 | .88/.10 | .87/.10 | .88/.31 |
| | DP10-10 | .97/.97 | .97/.97 | .98/.98 | .98/.98 | .97/.97 | .99/.94 |
| SVHN | D5-5 | .30/.22 | .28/.08 | .20/.08 | .20/.08 | .40/.20 | .28/.16 |
| | D9-1 | .60/.07 | .35/.07 | .67/.07 | .58/.07 | .61/.07 | .26/.10 |
| | DP10-10 | .81/.80 | .50/.50 | .20/.20 | .84/.84 | .82/.79 | .39/.29 |

DP10-10, D5-5:  50%  75%  100%

D9-1:  90%  95%  100%

**IMM is effective for all SLTs but unfeasible in practice.** As we can see from Tab. 4, wtIMM clearly outperforms all other models compared in Tab. 3. Especially for the D5-5 type SLTs, a modest incremental learning quality is attained, which is however quite far away from the baseline accuracy, even for MNIST-derived SLTs. This is in contrast to the results reported in Lee et al. (2017) for MNIST: we attribute this discrepancy to the application-oriented model selection procedure using only $D_1$ that we perform. In contrast, in Lee et al. (2017), a model with 800/800/800 neurons, for which good results on MNIST are well-established, is chosen beforehand, thus arguably making implicit use of $D_2$. A significant problem of IMM is the determination of the balancing parameter $\alpha$, exemplarily illustrated in Fig. 6. Our results show that the optimal value cannot simply be guessed from the relative sizes of $D_1$ and $D_2$, as it is done in Lee et al. (2017), but must be determined by cross-validation, thereby requiring knowledge of $D_1$ (violates constraints). Apart from these conceptual issues, we find that the repeated calculation of the Fisher matrices is quite time and memory-consuming (>4h and >8GB), to the point that the treatment of SLTs from certain datasets becomes impossible even on high-end machine/GPU combinations when using complex models. This is why we can evaluate IMM only for a few datasets. It is possible that this is an artifact of the TensorFlow implementation, but in the present state IMM nevertheless violates not one but two application constraints from Sec. 1.2. Fig. 7 and Fig. 8 give a visual impression of training an IMM model on D9-1 and D5-5 type SLTs, again illustrating basic feasibility, but also the variability of the "tuning curves" we use to determine the optimal balancing parameter $\alpha$.

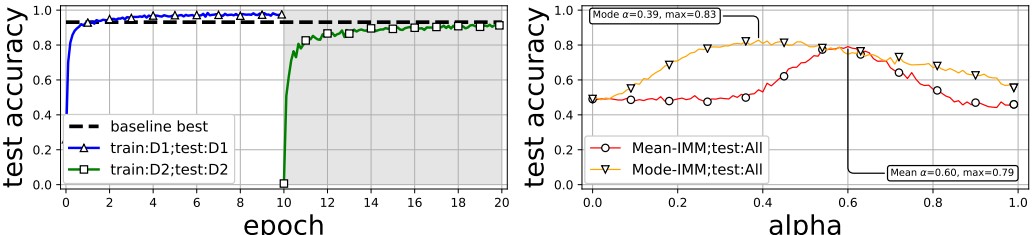

Figure 6: Accuracy measurements of best IMM model on SLT D5-5f for Devanagari dataset. On the left-hand side, the blue curve ($\triangle$) measures the accuracy of the first DNN trained on $D_1$, the green curve ($\square$) the accuracy of the second DNN trained on $D_2$. Additionally, the baseline (dashed line) is delineated. The right-hand side shows the tested accuracy on $D_1 \cup D_2$ of the merged DNN as a function of $\alpha$, both for mean-IMM (red $\bigcirc$) and the mode-IMM (orange $\triangledown$) variants. See also Appendix B for 2D plots.

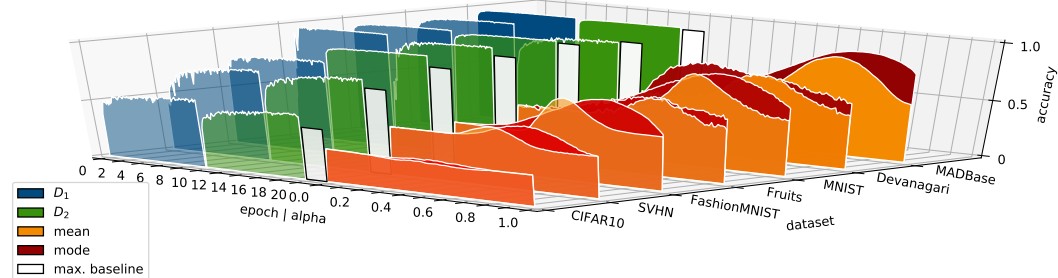

Figure 7: Best wtIMM experiments for SLT D5-5b constructed from datasets we were able to test. The blue surfaces (epochs 0-10) represent the test accuracy during training on $D_1$, the green surfaces the test accuracy on $D_2$ during training on $D_2$ (epochs 10-20). The white bars in the middle represent baseline accuracy, whereas the right part shows accuracies on $D_1 \cup D_2$ for different $\alpha$ values, computed for mean-IMM (orange surfaces) and mode-IMM (red surfaces). See also Appendix B for 2D plots.

## 5 CONCLUSIONS

The primary conclusion from the results in Sec. 4 is that CF still represents a major problem when training DNNs. This is particularly true if DNN training happens under application constraints as outlined in Sec. 1.2. Some of these constraints may be relaxed depending on the concrete application: if some prior knowledge about future sub-task exists, it can be used to simplify model selection

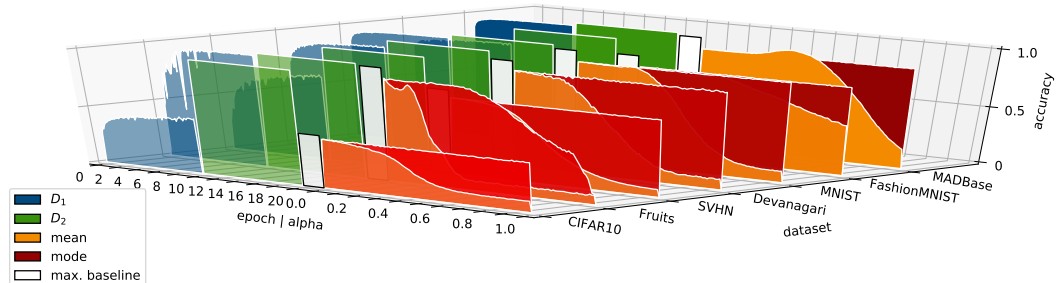

Figure 8: Best wtIMM experiments for the tested datasets for SLT D9-1c, to be read as Fig. 7. See also Appendix B for 2D plots.

Table 4: Summary of incremental learning quality $q_{\vec{p}^*}$, see Alg. 1, for the IMM model, evaluated on SLTs of type D9-1, D5-5 (DP10-10 is omitted because near-perfect performance was always attained). For aggregating results over SLTs of the same type, the minimal value of $q_{\vec{p}^*}$ (the best) is taken, as the presence of CF is indicated by a single occurrence of it in any SLT of the same type. To be interpreted as Tab. 3.

| | SLT | CIFAR10 | | Devanagari | | F_MNIST | | MADBase | | MNIST | | SVHN | |
| | Model | D5-5 | D9-1 | D5-5 | D9-1 | D5-5 | D9-1 | D5-5 | D9-1 | D5-5 | D9-1 | D5-5 | D9-1 |
|---|---|---|---|---|---|---|---|---|---|---|---|---|---|
| wtIMM | mode | .31 | .43 | .73 | .85 | .70 | .78 | .91 | .91 | .84 | .87 | .56 | .60 |
| | mean | .30 | .43 | .67 | .85 | .62 | .78 | .82 | .92 | .82 | .88 | .50 | .59 |

and improve results. If sufficient resources are available, a subset of previously seen data may be kept in memory and thus allow a "best" type evaluation/stopping criterion for re-training, see Alg. 1.

Our evaluation approach is similar to Kemker et al. (2018), and we adopt some measures for CF proposed there. A difference is the setting of up to 10 sub-tasks, whereas we consider only two of them since we focus less on the degree but mainly on presence or absence of CF. Although comparable both in the number of tested models and benchmarks, Serra et al. (2018) uses a different evaluation methodology imposing softer constraints than ours, which is strongly focused on application scenarios. This is, to our mind, the reason why those results differ significantly from ours and underscores the need for a consensus of how to measure CF.

In general application scenarios without prior knowledge or extra resources, however, an essential conclusion we draw from Sec. 4 is that model selection must form an integral part of training a DNN on SLTs. Thus, a wrong choice of hyper-parameters based on $D_1$ can be disastrous for the remaining sub-tasks, which is why application scenarios require DNN variants that do not have extreme dependencies on hyper-parameters such as layer number and layer sizes.

Lastly, our findings indicate workarounds that would make EWC or IMM practicable in at least some application scenarios. If model selection is addressed, a small subset of $D_1$ may be kept in memory for both methods: to determine optimal values of $\alpha$ for IMM and to determine when to stop re-training for EWC. Fig. 6 shows that small changes to $\alpha$ do not dramatically impact final accuracy for IMM, and Fig. 4 indicates that accuracy loss as a function of re-training time is gradual in most cases for EWC. The inaccuracies introduced by using only a subset of $D_1$ would therefore not be very large for both algorithms.

To conclude, this study shows that the consideration of applied scenarios significantly changes the procedures to determine CF behavior, as well as the conclusions as to its presence in latest-generation DNN models. We propose and implement such a procedure, and as a consequence claim that CF is still very much of a problem for DNNs. More research, either on generic solutions, or on workarounds for specific situations, needs to be conducted before the CF problem can be said to be solved. A minor but important conclusion is that results obtained on permutation-type SLTs should be treated with caution in future studies on CF.

ACKNOWLEDGMENTS

We gratefully acknowledge the support of NVIDIA Corporation with the donation of the Titan Xp GPU used for this research.

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

# A  ADDITIONAL EVALUATION METRICS

Table 5: Results of Tab. 3, using the measure $\Omega_{all}$ from Kemker & Kanan (2017). This is achieved by dividing the "best" measure from Tab. 3 by the baseline performance. Each table entry contains two numbers: the baseline performance and $\Omega_{all}$, and cell coloring (indicating presence or absence of CF) is performed based on $\Omega_{all}$. The overall picture is similar to the one from Tab. 3, as indicated by the cell coloring. A notable exception is the performance of the CONV and D-CONV models on the SVHN dataset, where $\Omega_{all}$ shows an increase, but we do not consider this significant since the already the baseline performance is at chance level here. That is, this problem is too hard for the simple architectures we use, in which case a small fluctuation due to initial conditions will exceed baseline performance. We therefore conclude that $\Omega_{all}$ is an important measure whenever baseline performance is better than random, in which case is it not meaningful. On the other hand, our measure works well for random baselines but is less insightful for the opposite case (as the presence of CF is not immediately observable from the raw performances. A combination of both measures might be interesting to cover both cases.

| DS | Model / SLT | FC | D-FC | CONV | D-CONV | LWTA | EWC |
|---|---|---|---|---|---|---|---|
| CIFAR10 | D5-5 | .50/.59 | .41/0.63 | .49/.64 | .48/0.63 | .52/.60 | .35/.91 |
| CIFAR10 | D9-1 | .51/.88 | .42/0.89 | .51/.90 | .52/0.94 | .51/.88 | .44/.82 |
| CIFAR10 | DP10-10 | .53/1.02 | .43/1.01 | .54/.97 | .50/1.12 | .52/1.03 | .51/1.12 |
| Devanagari | D5-5 | .95/.51 | .90/0.51 | .98/.50 | .99/0.50 | .91/.12 | .88/.45 |
| Devanagari | D9-1 | .97/.89 | .95/0.88 | .99/.88 | .99/0.89 | .96/.89 | .99/.89 |
| Devanagari | DP10-10 | .97/1.01 | .97/1.00 | .12/8.11 | .99/1.00 | .96/1.01 | 1.0/1.0 |
| EMNIST | D5-5 | .99/.51 | .99/0.51 | .99/.50 | 1.0/0.50 | .99/.51 | .94/.38 |
| EMNIST | D9-1 | .99/.89 | .99/0.89 | 1.0/.89 | 1.0/0.89 | .99/0.89 | 1.0/.92 |
| EMNIST | DP10-10 | .99/1.0 | .99/1.00 | 1.0/1.0 | 1.0/1.00 | .99/1.0 | 1.0/1.0 |
| F-MNIST | D5-5 | .87/.53 | .87/0.52 | .90/.52 | .91/0.51 | .88/.53 | .93/.59 |
| F-MNIST | D9-1 | .88/.88 | .87/0.88 | .91/.89 | .91/0.89 | .88/.88 | .95/.89 |
| F-MNIST | DP10-10 | .89/1.0 | .88/1.00 | .91/1.01 | .92/1.00 | .89/1.01 | .95/1.0 |
| Fruits | D5-5 | .51/.62 | .96/0.48 | .78/.17 | .88/0.16 | 1.0/.28 | .63/.54 |
| Fruits | D9-1 | .52/.66 | 1.0/0.38 | .79/.18 | .99/0.23 | .99/.39 | .91/.60 |
| Fruits | DP10-10 | 1.0/1.0 | 1.0/1.00 | .75/1.19 | .80/1.20 | .99/.99 | .78/1.26 |
| MADBase | D5-5 | .99/.50 | .98/0.50 | .99/.50 | .99/0.50 | .98/.50 | .97/.41 |
| MADBase | D9-1 | .99/.90 | .99/0.92 | .99/.90 | .99/0.90 | .98/.95 | 1.0/.99 |
| MADBase | DP10-10 | .99/1.0 | .99/1.00 | .99/1.0 | .99/1.00 | .98/1.0 | 1.0/1.0 |
| MNIST | D5-5 | .98/.51 | .97/0.51 | .99/.49 | .99/0.49 | .95/.10 | .94/.53 |
| MNIST | D9-1 | .98/.90 | .98/0.90 | .99/.88 | .99/0.88 | .98/.88 | 1.0/.99 |
| MNIST | DP10-10 | .99/1.0 | .98/1.00 | .99/1.0 | .99/1.00 | .98/1.0 | 1.0/1.0 |
| NotMNIST | D5-5 | .96/.51 | .96/0.51 | .97/.51 | .97/0.51 | .96/.51 | .99/.58 |
| NotMNIST | D9-1 | .97/.90 | .96/0.90 | .98/.90 | .98/0.90 | .97/.90 | 1.0/.88 |
| NotMNIST | DP10-10 | .97/1.0 | .97/1.00 | .98/1.0 | .98/1.00 | .97/1.0 | .99/1.0 |
| SVHN | D5-5 | .74/.40 | .37/0.76 | .20/.99 | .20/1.00 | .77/.52 | .30/.93 |
| SVHN | D9-1 | .75/.79 | .41/0.86 | .20/3.31 | .20/2.89 | .77/.79 | .30/.87 |
| SVHN | DP10-10 | .80/1.02 | .52/0.98 | .20/1.0 | .20/4.25 | .80/1.02 | .44/.89 |

90%    95%    100%

# B    SELECTED EXPERIMENTAL RESULTS AS 2D PLOTS

Here, we present the best results of all algorithms on the MNIST, EMNIST and Devanagari datasets (according to the "best" criterion) for the D9-1b SLT, and the best EWC results on the D5-5d SLT (qualitatively identical to the other D5-5 type SLTs). Such 2D representations of some experimental results, to be just as Fig. 4, may give more clear insights into the details of each experiment.

Here we can observe CF behavior for all algorithms except EWC and IMM for D9-1b. We can infer that there was no discernible dependency between the occurrence of CF and particular hyper-parameter settings (number and size of layers, in particular) since these are already the best experiments for each algorithm and dataset: if these show CF, this means that non of the settings we sampled were able to prevent CF. EWC shows clear CF for the Devanagari dataset, but might conceivably do better on EMNIST given a little more time for learning $D_2$ (this will be investigated). For D5-5d, clear CF occurs even for EWC. IMM does not exhibit CF for D9-1b (at enormous computations cost, though), and we observe that the value for the balancing parameter cannot simply be set to 0.9 respectively 0.1, as it has its argmax elsewhere.

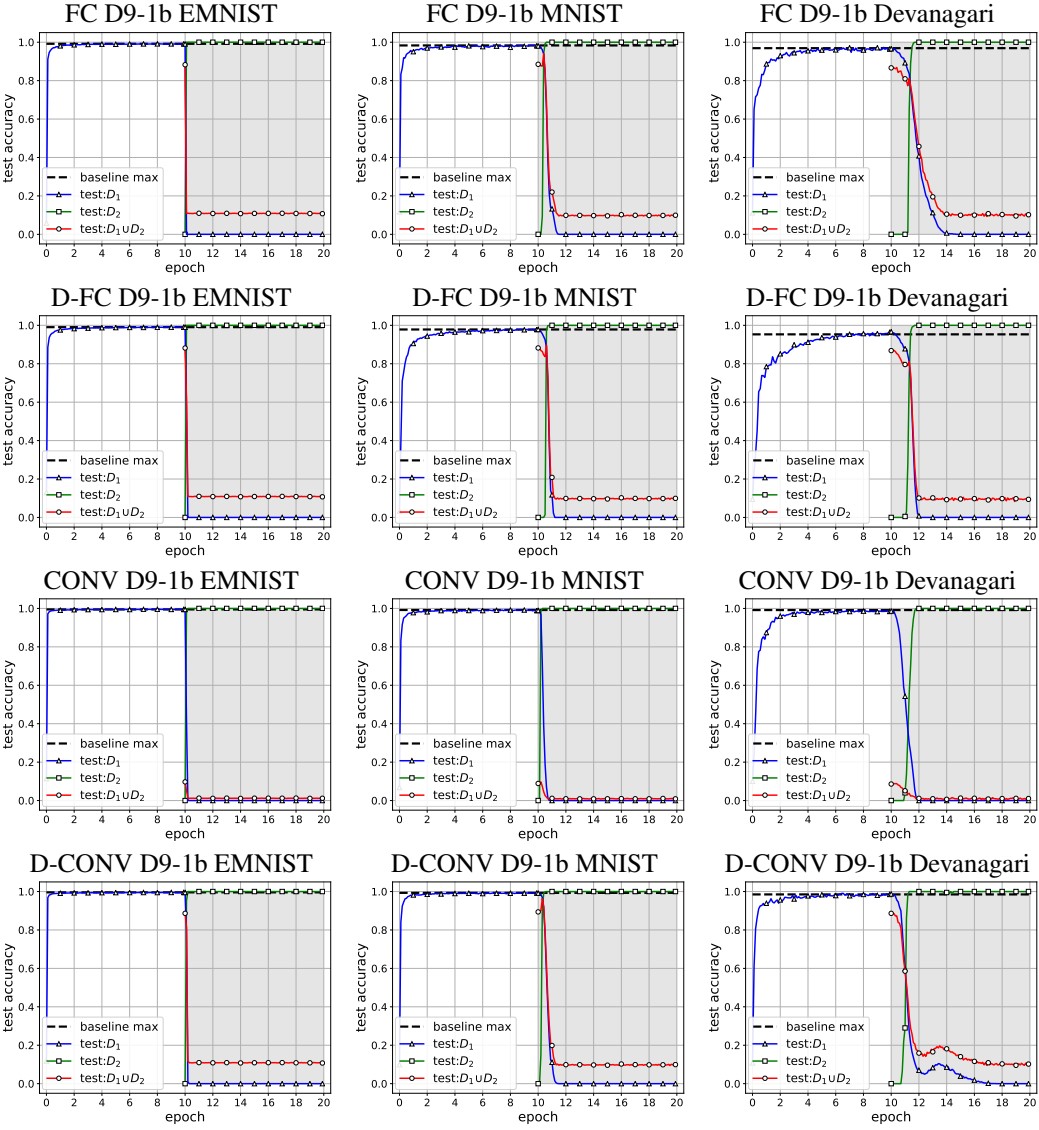

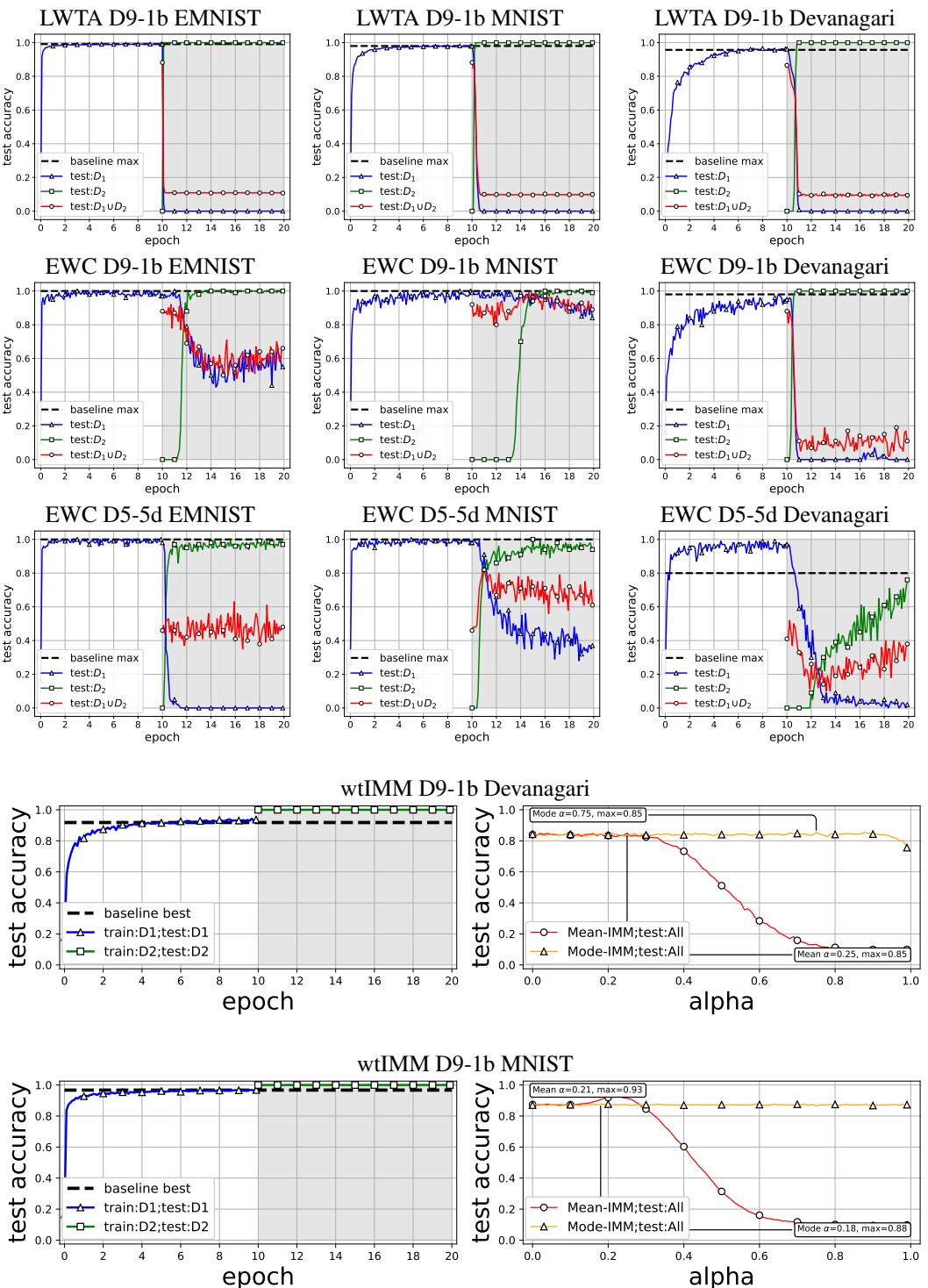

