# OpenReview forum: "A comprehensive, application-oriented study of catastrophic forgetting in DNNs"
_ICLR.cc/2019/Conference_

### Official Review · AnonReviewer1 · 2018-11-02
**A large and interesting analysis of CF in DNNs**

**Rating:** 7
**Confidence:** 3

**Review:**

# [Updated after author response]
Thank you for your response. I am happy to see the updated paper. In particular, the added item in section 1.3 highlights where the novelty of the paper lies, and as a consequence, I think the significance of the paper is increased. Furthermore, the clarity of the paper has increased.

In its current form, I think the paper would be a valuable input to the deep learning community, highlighting an important issue (CF) for neural networks. I have therefore increased my score.

------------------------------------------

# Summary
The authors present an empirical study of catastrophic forgetting (CF) in deep neural networks. Eight models are tested against nine datasets with 10 classes each but a varying number of samples. The authors construct a number of sequential learning tasks to test the model performances in different scenarios. The main conclusion is that CF is still a problem in all models, despite claims in other papers.

# Quality
The paper shows healthy criticism of the methods used to evaluate CF in previous works. I very much like this.

While I like the different experimental set-ups and the attention to realistic scenarios outlined in section 1.2, I find the analysis of the experiments somewhat superficial. The accuracies of each model for each task and dataset are reported, but there is little insight into what causes CF. For instance, do some choices of hyperparameters consistently cause a higher/lower degree of CF across models? I also think the metrics proposed by Kemker et al. (2018) are more informative than just reporting the last and best accuracy, and that including these metrics would improve the quality of the paper.

# Clarity
The paper is generally clearly written and distinct paragraphs are often highlighted, which makes reading and getting an overview much easier. In particular, I like the summary given in sections 1.3 and 1.4.

Section 2.4 describing the experimental setup could be clearer. It takes a bit of time to decipher Table 2, and it would have been good with a few short comments on what the different types of tasks (D5-5, D9-1, DP10-10) will tell us about the model performances. E.g. what do you expect to see from the experiments of D5-5 that is not covered by D9-1 and vice versa? And why are the number of tasks in each category so different (8 vs 3 vs 1)?

I am not a huge fan of 3D plots, and I don't think they do anything good in section 4. The perspective can make it tricky to compare models, and the different graphs overshadow each other. I would prefer 2D plots in the supplementary, with a few representative ones shown in the main paper. I would also experiment with turning Table 3 into a heat map.

# Originality
To my knowledge, the paper presents the largest evaluation of CF in terms of evaluated datasets. Kemker et al. (2018) conduct a somewhat similar experiment using fewer datasets, but a larger number of classes, which makes the CF even clearer. I think it would be good to cite this paper and briefly discuss it in connection with the current work.

# Significance
The paper is mostly a report of the outcome of a substantial experiment on CF, showing that all tested models suffer from CF to some extent. While this is interesting and useful to know, there is not much to learn in terms of what can cause or prevent CF in DNNs. The paper's significance lies in showing that CF is still a problem, but there is room for improvement in the analysis of the outcome of the experiments.

# Other notes
The first sentence of the second paragraph in section 5 seems to be missing something.

# References
Kemker, R., McClure, M., Abitino, A., Hayes, T., & Kanan, C. (2018). In AAAI Conference on Artificial Intelligence. https://aaai.org/ocs/index.php/AAAI/AAAI18/paper/view/16410

---

### Official Review · AnonReviewer3 · 2018-11-03
**An empirical study of CF, but more recent methods could have been also added for the study**

**Rating:** 6
**Confidence:** 5

**Review:**

Thanks for the updates and rebuttals from the authors.

I now think including the results for HAT may not be essential for the current version of the paper. I now understand better about the main point of the paper - providing a different setting for evaluating algorithms for combatting CF, and it seems the widespread framework may not accurately reflect all aspects of the CF problems.

I think showing the results for only 2 tasks are fine for other settings except for DP10-10 setting, since most of them already show CF in the given framework for 2 tasks. Maybe only for DP10-10, the authors can run multiple tasks setting, to confirm their claims about the permuted datasets. (but, I believe the vanilla FC model should show CF for multiple permuted tasks.)

I have increased my rating to "6: Marginally above acceptance threshold" - it could have been much better to at least give some hints to overcome the CF for the proposed setting, but I guess giving extensive experimental comparisons could be valuable for a publication.

=====================
Summary:

The paper evaluates several recent methods regarding catastrophic forgetting with some stricter application scenarios taken into account. They argue that most methods, including EWC and IMM, are prone to CF, which is against the argument of the original paper.

Pro:
- Extensive study on several datasets, scenarios give some intuition and feeling about the CF phenomenon.

Con:
- There are some more recent baselines., e.g., Joan Serrà, Dídac Surís, Marius Miron, Alexandros Karatzoglou, "Overcoming catastrophic forgetting with hard attention to the task" ICML2018, and it would be interesting to see the performance of those as well.
- The authors say that the permutation based data set may not be useful. But, their experiments are only on two tasks, while many work in literature involves much larger number of tasks, sometimes up to 50. So, I am not sure whether the paper's conclusion that the permutation-based SLT should not be used since it's only based on small number of tasks.
- While the empirical findings seem useful, it would have been nicer to propose some new method that can get around the issues presented in the paper.

---

### Official Review · AnonReviewer2 · 2018-11-05
**Interesting topic of limited novelty**

**Rating:** 5
**Confidence:** 4

**Review:**

The paper presents a study of the application of some well known methods on 9 datasets focusing on the issue of catastrophic forgetting when considering a sequential learning task in them.  In general the presentation of concepts and results is a bit problematic and unclear. Comments, such that the paper presents ' a novel training and model selection paradigm for incremental learning in DNNs ' is not justified.  A better description of the results, e.g., in Table 3 should be presented, as well a better linking with the findings; a better structure of the latter would also be required to improve consistency of them. Improving these could make the paper candidate for a poster presentation.

---

### Meta-Review · Area_Chair1 · 2018-12-13
**Catastrophic forgetting: not dead yet**

**Confidence:** 3
**Recommendation:** Accept (Poster)

**Metareview:**

This paper has two main contributions. The first is that it proposes a specific framework for measuring catastrophic forgetting in deep neural networks that incorporates three application-oriented constraints: (1) a low memory footprint, which implies that data from prior tasks cannot be retained; (2) causality, meaning that data from future tasks cannot be used in any way, including hyperparameter optimization and model selection; and (3) update complexity for new tasks that is moderate and also independent of the number of previously learned tasks, which precludes replay strategies. The second contribution is an extensive study of catastrophic forgetting, using different sequential learning tasks derived from 9 different datasets and examining 7 different models. The key conclusions from the study are that (1) permutation-based tasks are comparatively easy and should not be relied on to measure catastrophic forgetting; (2) with the application-oriented contraints in effect, all of the examined models suffer from catastrophic forgetting (a result that is contrary to a number of other recent papers); (3) elastic weight consolidation provides some protection against catastrophic forgetting for simple sequential learning tasks, but fails for more complex tasks; and (4) IMM is effective, but only if causality is violated in the selection of the IMM balancing parameter. The reviewer scores place this paper close to the decision boundary. The most negative reviewer (R2) had concerns about the novelty of the framework and its application-oriented constraints. The authors contend that recent papers on catastrophic forgetting fail to apply these quite natural constraints, leading to the deceptive conclusion that catastrophic forgetting may not be as big of a problem as it once was. The AC read a number of the papers mentioned by the authors and agrees with them: these constraints have been, at least at times, ignored in the literature, and they shouldn't be ignored. The other two reviewers appreciated the scope and rigor of the empirical study. On the balance, the AC thinks this is an important contribution and that it should appear at ICLR.